# Virulence and transmission vary between Usutu virus lineages in *Culex pipiens*

**Maxime Prat[1,2], Mélanie Jeanneau[2], Ignace Rakotoarivony[2], Maxime Duhayon[2], Yannick Simonin[3], Giovanni Savini[4], Pierrick Labbé[1], Haoues Alout[2]***

**1** Institut des Sciences de l'Evolution de Montpellier, Université de Montpellier-CNRS-IRD, Montpellier, France, **2** UMR ASTRE, Univ Montpellier, INRAE-CIRAD, Montpellier, France, **3** Pathogenesis and Control of Chronic Infections, Université de Montpellier-INSERM-EFS, Montpellier, France, **4** OIE Reference Centre for West Nile Disease, Istituto Zooprofilattico Sperimentale "G. Caporale", Teramo, Italy

* haoues.alout@inrae.fr; haoues.alout@cirad.fr

**Data Availability Statement:** Data and scripts are available at https://zenodo.org/records/11093366?oken=eyJhbGciOiJIUzUxMiIsImIhdCl6MTcxNDQ4NDY2MiwiZXhwljoxNzI1MTQ4Nzk5fQ.eyJpZCl

## Abstract

Usutu virus (USUV) is a zoonotic arbovirus infecting mainly wild birds. It is transmitted by ornithophilic mosquitoes, mainly of the genus *Culex* from birds to birds and to several vertebrate dead-end hosts. Several USUV lineages, differing in their virulence have emerged in the last decades and now co-circulate in Europe, impacting human populations. However, their relative transmission and effects on their mosquito vectors is still not known. We thus compared the vector competence and survival of *Culex pipiens* mosquitoes experimentally infected with two distinct USUV lineages, EU2 and EU3, that are known to differ in their virulence and replication in vertebrate hosts. Infection rate was variable among blood feeding assays but variations between EU2 and EU3 lineages were consistent suggesting that *Culex pipiens* was equally susceptible to infection by both lineages. However, EU3 viral load increased with viral titer in the blood meal while EU2 viral load was high at all titers which suggest a greater replication of EU2 than EU3 in mosquito. While their relative transmission efficiencies are similar, at least at low blood meal titer, positive correlation between transmission and blood meal titer was observed for EU3 only. Contrary to published results in vertebrates, EU3 induced a higher mortality to mosquitoes (i.e. virulence) than EU2 whatever the blood meal titer. Therefore, we found evidence of lineage-specific differences in vectorial capacity and virulence to both the vector and vertebrate host which lead to balanced propagation of both viral lineages. These results highlight the need to decipher the interactions between vectors, vertebrate hosts, and the diversity of arbovirus lineages to fully understand transmission dynamics.

## Author summary

Usutu virus is a zoonotic arbovirus that emerged in the last decades in Europe, impacting wild bird and human populations. Several co-circulating USUV lineages differ in their virulence to vertebrate hosts but their relative transmission and effects on their mosquito vectors is still not known. We compared the vector competence and survival of *Culex pipiens* mosquitoes experimentally infected with two distinct USUV lineages. We showed

6IjJiYjhhZThjLTE4NTgtNGQyZS1iZDBkLThIMTllZ
TRmYTgzMSIsImRhdGEiOnt9LCJyYW5kb20iOiI0
YTgyNDYzYThmYmNjNzRhMTdjNjJkOTkzMGE4
NzVmNCJ9.tXk_mfDPknpzN_n0lBrg4DCMMqggEv
JGmCdPB8QJ5i7TzLeAHX7nG_Swi-9F2
0fxMFuK9kM1Xj8SK9lv8A-UEg.

**Funding:** This work has been funded in part by the ANR PRC "ArchR" ANR-20-CE34-0007 (2021-2024) to PL and HA. PhD fellowship to MP has been funded by the ANR PRC "ArchR" ANR-20-CE34-0007 and the Animal Health department of INRAE. The funders had no role in study design, data collection and analysis, decision to publish, or preparation of the manuscript.

**Competing interests:** The authors have declared that no competing interests exist.

that their relative transmission efficiencies are similar but EU3 was more virulent to mosquitoes than EU2. In addition to published results, we revealed lineage-specific differences in vectorial capacity and virulence to both the vector and the vertebrate host suggesting that propagation of both viral lineages might be balanced. Our study provides new insights into the transmission of an emerging arbovirus and highlights the need to investigate the interactions between vectors, vertebrate hosts, and the diversity of arbovirus lineages to fully understand transmission dynamics.

## Introduction

Usutu Virus (USUV) is a mosquito-borne flavivirus (family: *Flaviviridae*, genus: *flavivirus*) and is part of the Japanese encephalitis serocomplex with other phylogenetically related virus like Yellow Fever virus (YFV), Japanese Encephalitis virus (JEV) and West Nile virus (WNV). It was first identified in 1959 [1] along the Usutu River in Swaziland. Like WNV, USUV is maintained in a transmission cycle involving wild birds and ornithophilic mosquitoes, mainly of the genus *Culex* [2]. There have been recurrent outbreaks of USUV among birds with a few instances of spillover into humans and other mammals in several European countries since its emergence in 1996 in Italy [3,4]. These outbreaks were characterized by high mortality rates in passeriform and strigiform birds, with some species, such as blackbirds (*Turdus merula*), being particularly affected. Infections in other animals have been reported, including bats of the genus *Pipistrellus* [5], dogs [6,7], ruminants [8], wild boar [9], shrews and rodents [10], horses [11,12,13], and humans [2,4]. However, infections in these non-avian hosts are likely not transmissive (dead-end hosts) as for West Nile virus [14].

Flavivirus RNA genomes experience high mutation rates, which may facilitate rapid adaptation to new environments with potentially new hosts and new vectors [15]. As population sizes increase and the virus becomes more widespread, there is a greater likelihood of fitness-enhancing mutations furthering spread within and across continents. Such adaptation may directly and indirectly affect patterns of viral transmission, virulence, and host specificity. Phylogenetic studies have shown that USUV isolates can be divided into eight lineages: five European lineages (EU1 to EU5) and three African lineages (AF1 to AF3) [16]. Many of these lineages have been found co-circulating in Europe (i.e., EU1, EU2, EU3, EU5, AF2, and AF3). EU2 is the most detected lineage in mosquitoes, birds, and humans in central and southern Europe, while EU3 is the most prevalent in northern Europe [4,17]. Analyses of viral pathogenesis in mice and human cell culture revealed significant variation in virulence among-lineage: EU2 lineage appeared to be the most virulent, as it was more pathogenic to mice and replicated more efficiently and persistently in human neuronal cells [18] while AF2 and EU3 lineages appeared to be the least virulent [18,19].

Vectorial capacity is defined as the transmission potential of a vector population and depends on both ecological (vector density $m$, daily host biting rate $a$, daily survival rate $p$) and biological traits (vector competence $c'$, Extrinsic Incubation Period or EIP $n$), as summarized in Eq 1 [20,21]:

$$C = \frac{m a_1 a_2 p^n c'}{-\ln p} \qquad (1)$$

**Eq 1. Vectorial capacity equation for multi-host transmission.** Parameters are vector density $m$, daily host biting rate $a$ ($a_1$ is the biting rate on one host; $a_2$ on another host), daily survival rate $p$, vector competence $c'$, and Extrinsic Incubation Period $n$.

These traits are themselves affected by environmental factors (temperature, humidity, chemical exposure) modulating vector-borne transmission. Assessing vectorial capacity is critical for understanding transmission dynamics and, managing and controlling arbovirus transmission.

Vector competence defines the intrinsic ability of mosquitoes to transmit pathogens—from feeding on an infected host to transmission to a new host, thus including within-vector viral replication and dissemination up to the salivary glands. Competent mosquitoes will acquire virus from an infected host and later transmit the virus to other susceptible birds, or other animals, including humans. Entomological surveillances have reported USUV infections in several ornithophilic mosquitoes, mainly of the *Culex* genus but also in *Culiseta annulata*, *Anopheles maculipenis*, *Aedes spp*, *Mansonia spp*, and *Ochlerotatus spp* [3]. However, their role as vector for USUV has only been demonstrated in four *Culex* species: *Cx. neavi*, *Cx. pipiens* (biotypes *pipiens* and *molestus*), *Cx. quinquefasciatus*, and *Cx. torrentium* [22,23].

Environmental, genetic, symbiotic, and ecological factors may affect the likelihood of disease transmission by vectors, but genetic factors in the arboviruses themselves may also play a significant role. Given the significant pathogenic difference on vertebrate hosts among USUV lineages, it is important to consider whether similar difference is observed in how these lineages interact with and are transmitted by their vectors. We considered this question within the context of the virulence-transmission tradeoff hypothesis. The hypothesis posits that while highly replicating viruses may be transmitted more readily, they risk killing the host before transmission can occur. Thus, natural selection might favor viruses that balance high transmission and host mortality, resulting in intermediate levels of virulence [24–26]. Evidence of virulence-transmission tradeoffs has been suggested for rodent malaria [27], Schistosoma [28], and myxoma [27], although more data is needed to confirm their generality [28]. There is some evidence that mosquito-borne pathogens can reduce the longevity of infected mosquitoes [29] and, therefore, their vectorial capacity, which calls into question whether the virulence-transmission tradeoff is also important in the vector. Additionally, because the transmission cycle of arboviruses involves the infection of arthropod and vertebrate hosts, arboviruses may face an additional tradeoff if adaptation to one of the hosts negatively affects adaptation to the other [30].

Here, we investigated the relative vectorial capacity for two USUV lineages (EU2 and EU3), which differ significantly in their virulence in the vertebrate host. Throughout the 14 days of extrinsic incubation period [31], we measured daily mortality, viral load, and infection prevalence in diverse tissues of *Cx. pipiens* biotype *pipiens* mosquitoes (hereafter named *Cx. p. pipiens*) following infectious blood meals containing high or low viral titers to better characterize *Culex*-USUV interactions. We tested a 100-fold variation of viral titer reflecting the variation of replication of EU2 and EU3 linages in vertebrate host [18]. Our analyses allowed us to test for differences between lineages in transmission and mortality as well as the relationship between these traits within the mosquito and compared to patterns observed in the vertebrate host.

## Materials and methods

### Mosquito lines

*Culex p. pipiens* larvae were collected in September 2021 from ponds in a protected natural area of the Camargue at La Tour du Valat (https://tourduvalat.org/, coordinate: 43.509297, 4.668958) and maintained in the insectarium. PCR of the polymorphic Cq11 microsatellite [32] was used to confirm that mosquitoes belong to the biotype *pipiens*. The *C. p. pipiens* larvae and adults were raised in a secured insectarium (Baillarguet insectarium platform,

Montpellier) at 27±1˚C, 65±8% relative humidity, and a 14h light/10h dark photoperiod. After egg-hatching, approximately 500 larvae per tray were randomly dispatched into three to five plastic trays containing 1 liter of tap water and fed with Tetramin fish granules. At emergence, adults were fed *ad libitum* with a 5% honey solution and maintained in 30x30x30 cm cages. The day before blood feeding, mosquitoes were starved by removing the honey solution.

### Viral strains

Two USUV isolates were used, the first belonging to the Europe 2 lineage (EU2, Accession Number [AN]: MT784899) and the second to the EU3 lineage (AN: KX601690) [33]. They were isolated after three passages on VERO cells (ATCC CCL-81) and one on C6/36 cells (ATCC CRL-1660). All virus stocks were then amplified on Vero cells in MEM media, and the supernatants stored at -80˚C until experimental infections. EU2 and EU3 virus stocks were titrated to $10^{7.2}$ TCID$_{50}$/mL and $10^{6.8}$ TCID$_{50}$/ml, respectively (Tissue Culture Infectious Dose 50), using the SpearmanKärber method [34].

### Infection by membrane feeding assay

Infectious blood feedings were carried out with 5- to 8-day-old females who had not previously taken a blood meal. The day before the infectious blood meal, small cages (15x15x15 cm) containing approximately 200 females were placed in climatic chambers in the BSL3 laboratory (BioSecurity Level 3) with the same constant environmental conditions as the insectarium but no access to the honey solution. On the day of infection, 2 ml of infectious blood was prepared by mixing 1.5 ml of heparinized turkey blood, 0.5 mM ATP as a phagostimulant [35], and 500 μl of MEM containing virus at a final titer of $10^4$ TCID$_{50}$/ml (low titer) or $10^6$ TCID$_{50}$/ml (high titer). Mosquitoes were allowed to feed on infectious blood using a Hemotek system at 37˚C for 3 hours in darkness. Control females were treated in the same manner but fed with blood containing ATP and MEM but lacking virus. After blood feeding, 500μl of infectious blood was stored at -80˚C, and mosquitoes were anesthetized using $CO_2$ to sort out the fully engorged females. They were kept individually for 14 days in *Drosophila* tubes containing a moistened foam and topped with a veil on which sat a cotton ball soaked in a 5% honey solution that was renewed every 2–3 days. At the end of this period, saliva was collected from live mosquitoes who were first anesthetized to remove the wings and legs and then placed with their proboscis inserted into 20 μL tips filled with 10 μL of MEM + 10% Fetal Bovine Serum (FBS). Saliva was collected for 30 minutes and stored at -80˚C. Then, the thorax, head, and midgut of each female were dissected and stored separately at -80˚C. Experimental infections were repeated three times for each USUV lineage and blood titer, resulting in 12 independent experimental infections.

### Survival analysis

Every day over the 14-day course of infection, dead mosquito females were recorded daily and stored at -80˚C. Dead and surviving females were analyzed for USUV infection (see method below). Note that females that died at day 1 were removed from the analysis to avoid virus detection from undigested or partially digested blood in the midgut.

### Vector competence analysis

Total RNA was extracted using the KingFisher Flex automat and the NucleoMag RNA isolation kit (Macherey-Nagel) according to the manufacturer protocols and stored at -80˚C. The detection and quantification of USUV viral genome was carried out by RT-qPCR as described

in Nikolay *et al.* [36] using the NEB Luna Universal One-step RT-qPCR kit and the primer set: USUV-Forward (5'-AAAAATGTACGCGGATGACACA-3') and USUV-Reverse (5'-TTTGG CCTCGTTGTCAAGATC-3'). The samples were analyzed in duplicate concomitantly with a negative control and serial dilutions of USUV RNA extracted from USUV virus stock as standard to quantify individual viral load. The limit of detection has been defined as the Ct value of the last standard dilution with reproducible detection (i.e. all replicates for a dilution are positives, see S1 Fig). Vector competence was assessed based on several parameters: i) the prevalence of infection is the proportion of infected tissues among mosquitoes alive at 14 dpi (days post-infection), and ii) the viral load is the number of virus genome copies per mL of extracted RNA in infected tissues. For a fine characterization of vector competence, we calculated the transmission efficiency (TE: the proportion of mosquitoes with infected saliva among all blood-fed mosquitoes), the dissemination efficiency (DE: the proportion of mosquitoes with infected thoraces among all blood-fed mosquitoes), the infection efficiency (IE: the proportion of mosquitoes with infected midguts among all blood-fed mosquitoes), the dissemination rate (DR: the proportion of mosquitoes with infected thoraces among mosquitoes with infected midguts), and the transmission rate (TR: the proportion of mosquitoes with infected saliva among mosquitoes with infected thoraces). The infection status of dead mosquitoes was determined by the detection of virus in the whole body by RT-qPCR.

## Statistical analysis

We analyzed the outcome of USUV infections in *Cx. p. pipiens* using two parameters as response variables: i) the infection prevalence and ii) the viral load. We examined the effects of diverse explanatory variables: "lineage" (categorial variable with two levels: EU2 and EU3), "blood meal titer" (a categorical variable with two levels: low and high), "true blood meal titer" (viral titer in blood after blood-feeding, numerical variable), "wing size" (numerical variable) and "replicate" (categorical variable with six levels: A, B, C, D, E, and F).

All statistical analyses were conducted using the R software 4.2.0. We computed generalized linear mixed effect models using the *glmmTMB* package [37] with i) a binomial error structure for analyzing the prevalence and ii) a negative binomial error structure for the viral load. Maximal models included the variables "lineage", "blood meal titer", "true blood meal titer", "wing size" and all their interactions. The "replicate" variable was used as a random variable to account for the nested data structure, i.e., the correlations between individuals from the same feeding experiment, and was either selected or removed based on the lowest AIC. Failed virus quantification in one tissue was specified as NA (Non Available) and was not included in the analysis. The significance of variables and selection of the minimal model was assessed using the ANOVA procedure within the package *car* [38], which performs type III hypothesis likelihood ratio tests (LRT). Mean parameter estimates were computed and *post-hoc* tests carried out to assess the differences between estimates with corrections for multiple comparisons using the package *emmeans* [39]. Kaplan Meier survival analyses were carried out and compared with that of the control group. The differences between survival distributions were estimated using the asymptotic Log-Rank Test [40], and the Hazard Ratio (HR defined as the ratio of the risk of death) was estimated from the *riskRegression* package [41].

The relative vectorial capacity of EU3 over EU2 has been calculated using the estimated vector competence (c') and survival (p) from this study keeping all other parameters equals (m, a and n). After simplification, the equation of the relative vectorial capacity is:

$$\frac{C_{EU3}}{C_{EU2}} = \left(\frac{p_{EU3}}{p_{EU2}}\right)^n \times \frac{\ln p_{EU2}}{\ln p_{EU3}} \times \frac{c'_{EU3}}{c'_{EU2}} \tag{2}$$

**Eq 2. Equation of relative vectorial capacity.** The equation was simplified based on fixed values for the density (m), the biting rate (a) and the extrinsic incubation period (n = 14 days).

## Results

We examined the vector competence of *Cx. p. pipiens* for two USUV lineages (EU2 and EU3) provided in blood meals containing two distinct viral titers (referred as blood meal titers). We measured infection parameters and mortality after 14 days. Overall, 559 blood-fed females (188 with EU2, 236 with EU3, and 135 controls) from 3 replicates (*i.e.*, blood-feeding assays) per blood meal titer and USUV lineage were analyzed.

### Prevalence of USUV infection in diverse tissues

We first investigated the USUV prevalence in mosquito to analyze the different parameters over all tissues and then analyze the different tissues separately to characterize the vector competence by calculating the infection, dissemination, and transmission efficiencies (IE, DE, and TE, respectively; see *Vector competence analysis* in *Material and Method*). Considering all tissues together (S1 Table), the prevalence of infection was a significantly influenced by the "*bloodmeal titer*" ($\chi^2$ = 7.61; *p*-value = 0.0058) and the "*bloodmeal titer x lineage*" interaction ($\chi^2$ = 6.22; *p*-value = 0.012). Prevalence was higher when mosquitoes fed on blood meal containing a higher virus titer than the one fed with lower virus titer for EU3 but not for EU2. In addition, infection prevalence was significantly influenced by the "*bloodmeal titer x tissue*" interaction ($\chi^2$ = 10.54; *p*-value = 0.0051) suggesting a different impact of bloodmeal titer in the different tissues. The minimal model revealed a significant effect of "*Lineage x Bloodmeal titer x Wing size*" interaction indicating an influence of mosquito size on infection that depends on USUV lineage and on the bloodmeal titer.

Focusing on the different tissues, the statistical analyses showed: i) For IE (midguts), none of the variable had a significant effect on prevalence for EU2 and EU3 (see Table 1 and Fig 1A) (low titer: $IE_{EU2}$ = 0.24 ± 0.19 and $IE_{EU3}$ = 0.46 ± 0.25, *p*-value > 0.05; high titer: $IE_{EU2}$ = 0.45 ± 0.24 and $IE_{EU3}$ = 0.61 ± 0.23, *p*-value > 0.05; Fig 1A). ii). For DE (thoraces), none of the variables had a significant effect (Table 1 and Fig 1B; low titer: $DE_{EU2}$ = 0.14 ± 0.11 and $DE_{EU3}$ = 0.20 ± 0.14, *p*-value > 0.05; high titer: $DE_{EU2}$ = 0.38 ± 0.21 and $DE_{EU3}$ = 0.52 ± 0.22, *p*-value > 0.05, Fig 1B). iii). For TE (saliva), the "*Lineage x Blood meal titer*" interaction was significant ($\chi^2$ = 6.45; *p*-value = 0.011, Table 1 and Fig 1C), as well as the "*Lineage*" effect ($\chi^2$ = 4.83, *p*-value = 0.027). Difference in TE was not significant between EU2 and EU3 at low titer contrary to high titer (low titer: $TE_{EU2}$ = 0.07 ± 0.04 and $TE_{EU3}$ = 0.02 ± 0.02, *p*-value = 0.11) but significant at high titer: ($TE_{EU2}$ = 0.10 ± 0.05 and $TE_{EU3}$ = 0.25 ± 0.1, *p*-value = 0.03).

**Table 1. Statistical analyses of infection, dissemination and transmission efficiencies of USUV lineages 14 days post infection.**

| Variable | Infection efficiency (IE) | | | Dissemination efficiency (DE) | | | Transmission efficiency (TE) | | |
|---|---|---|---|---|---|---|---|---|---|
| | $\chi^2$ | d.f. | *p-value* | $\chi^2$ | d.f. | *p-value* | $\chi^2$ | d.f. | *p-value* |
| Lineage | 3.67 | 1 | 0.051 | 2.49 | 1 | 0.11 | **4.83** | **1** | **0.027** |
| Bloodmeal titer | 0.42 | 1 | 0.51 | 1.03 | 1 | 0.30 | 0.27 | 1 | 0.59 |
| Interaction | 0.19 | 1 | 0.65 | 0.08 | 1 | 0.77 | **6.45** | **1** | **0.011** |

Parameters selected in the minimal generalized linear mixed models are presented. $X^2$ is the Chi-square value of the likelihood ratio test and d.f. is the degree of freedom. The p-values are bolded when significant.

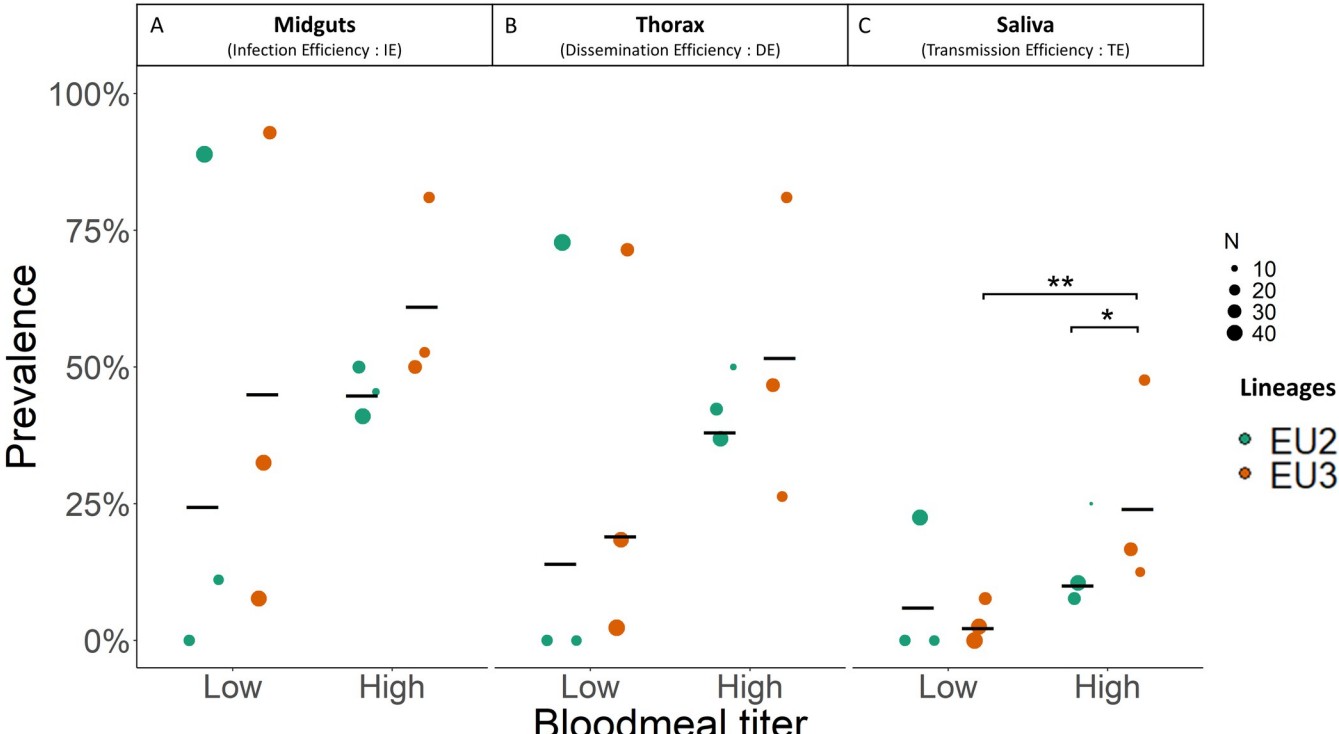

**Fig 1. Prevalence of USUV infection in different *Cx. p. pipiens* tissues.** The proportions of individual mosquitoes infected 14 dpi with EU2 (green points) and EU3 lineages (orange points), at low (left) and high (right) bloodmeal titer, are indicated for three body parts: A) midgut, B) thorax and C) saliva. Circles represent each replicate, with their diameter proportional to the corresponding sample size. Black bars represent the mean of the three replicates with overall sample sizes of 84 for EU2 and 109 for EU3 at low bloodmeal titer, and 76 for EU2 and 70 for EU3 at high bloodmeal titer. Only the significant differences are represented with stars (*: *p*-value < 0.05).

However, EU3 lineage present a significant dose-effect on TE (TE$_{EU3}$ low vs. TE$_{EU3}$ high, p-value = 0.004, Fig 1C).

Overall, the dissemination rate (DR, the proportion of infected thoraces among mosquitoes with infected midguts) and the transmission rate (TR, the proportion of mosquitoes with infected saliva among those with infected thoraces; Table 2 and Fig 2) were below 100%, indicating the presence of infection barriers in mosquitoes preventing USUV dissemination outside the midgut and in mosquito saliva. Neither USUV lineage nor blood meal titer significantly affected the DR (Fig 2A; low titer: DR$_{EU2}$ = 0.14 ± 0.11, DR$_{EU3}$ = 0.18 ± 0.13; *p*-value = 0.45; high titer: DR$_{EU2}$ = 0.37 ± 0.2 and DR$_{EU3}$ = 0.52 ± 0.2, *p*-value = 0.09; EU2 low vs high *p*-value = 0.31, EU3 low vs high *p*-value = 0.21). By contrast, the TR was significantly

**Table 2. Dissemination rate and transmission rates of USUV lineages by *Cx. p. pipiens*.**

| Blood meal titer | Low | | High | |
|---|---|---|---|---|
| USUV lineage | EU2 | EU3 | EU2 | EU3 |
| N infected midguts | 44 | 42 | 34 | 42 |
| N infected thorax | 32 | 28 | 30 | 36 |
| N infected saliva | 9 | 3 | 8 | 17 |
| Dissemination Rate (DR) | 0.14 +/- 0.11 | 0.18 +/- 0.13 | 0.36 +/- 0.2 | 0.51 +/- 0.21 |
| Transmission Rate (TR) | 0.06 +/- 0.04 | 0.02 +/- 0.016 | 0.09 +/- 0.04 | 0.23 +/- 0.09 |

Estimates from general linear mixed models are presented with standard error.

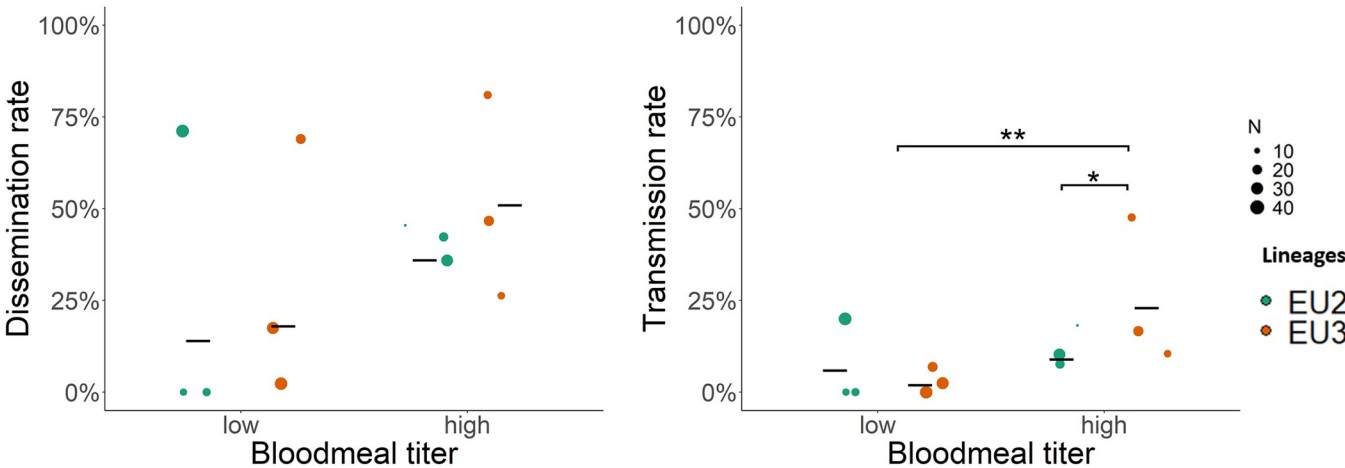

**Fig 2. Dissemination rate (DR) and transmission rate (TR) of EU2 and EU3 lineages in *Cx. p. pipiens*.** The mean proportions of mosquitoes with infected thoraces among those with infected midguts (DR, panel A) and of individual mosquitoes with infected saliva among those with infected thoraces (TR, panel B) are indicated for EU2 (green points) and EU3 lineages (orange points), at low (left) and high (right) viral titer. Circles represent each replicate, with their diameter proportional to the corresponding sample size. Black bars represent the mean of the three replicates. Sample sizes are presented in Table 2. Only the significant differences are represented with stars (*: *p*-value < 0.05).

lower for USUV EU3 at low titer but not different between titers for EU2 or between EU2 and EU3 at high titer (Fig 2B; low titer: $TR_{EU2}$ = 0.06 ± 0.04 and $TR_{EU3}$ = 0.02 ± 0.016, *p*-value = 0.11; high titer: $TR_{EU2}$ = 0.09 ± 0.04 and $TR_{EU3}$ = 0.23 ± 0.09, *p*-value = 0.02; EU2 low vs high *p*-value = 0.61, EU3 low vs high *p*-value = 0.004).

## Viral load in mosquito tissues

The analysis of infected females showed that viral load (VL, expressed in genome copies GE/mL) was generally lower for EU3 than EU2, but only at low titer and not in all tissues (Fig 3): the "*Lineage x Blood meal titer*" interaction was significant for VL in midguts and thoraces (respectively $\chi^2$ = 180.8; *p*-value < 0.0001 and $\chi^2$ = 139.68; *p*-value < 0.0001; Table 3). Post hoc analyses revealed that EU3 viral load was significantly lower than EU2 at low titer but not at high titer in both the midgut and thorax (Fig 3A: low titer: $VL_{EU2}$ = 1.15x$10^6$ ± 5.16x$10^6$ GE/ml and $VL_{EU3}$ = 2.79 ± 1.36 GE/ml, *p*-value < 0.0001; high titer: $VL_{EU2}$ =: 2.64x$10^6$ ± 1.38x$10^6$ GE/ml, $VL_{EU3}$ = 2.88x$10^6$ ± 1.37x$10^6$ GE/ml, *p*-value = 0.99; EU2 low vs high *p*-value = 0.6, EU3 low vs high *p*-value < 0.0001; Fig 3B: low titer: $VL_{EU2}$ = 8.62x$10^6$ ± 4.24x$10^6$ GE/ml and $VL_{EU3}$ = 33.4 ± 11.7 GE/ml, *p*-value < 0.0001; high titer: $VL_{EU2}$ = 3.07x$10^6$± 1.56x$10^6$ GE/ml, $VL_{EU3}$ = 1.75x$10^6$ ± 8.36x$10^5$ GE/ml, *p*-value = 0.85; EU2 low vs high *p*-value = 0.46, EU3 low vs high *p*-value < 0.0001). In saliva, viral load was much lower than in the other tissues (Fig 3C) and was slightly higher in mosquitoes that had taken a blood meal with a higher titer ($\chi^2$ = 5.88; *p*-value = 0.015). Post hoc analysis showed that there was only a slightly significant increase in EU2 viral load between titers (Fig 3C; low titer: $VL_{EU2}$ = 594.1 ± 416 GE/ml vs high titer: $VL_{EU2}$ =: 7.08x$10^3$± 5.26x$10^3$, *p*-value = 0.02). All other pairwise comparisons were not significant.

## Effect of USUV on mosquito survival

Mortality was analyzed in two ways. First, we compared survivorship between mosquitoes fed on infectious blood (EU2 or EU3) and control mosquitoes fed on non-infectious blood (Fig 4 and Table 4). For each blood meal titer (low and high), the Kaplan-Meier survival curves

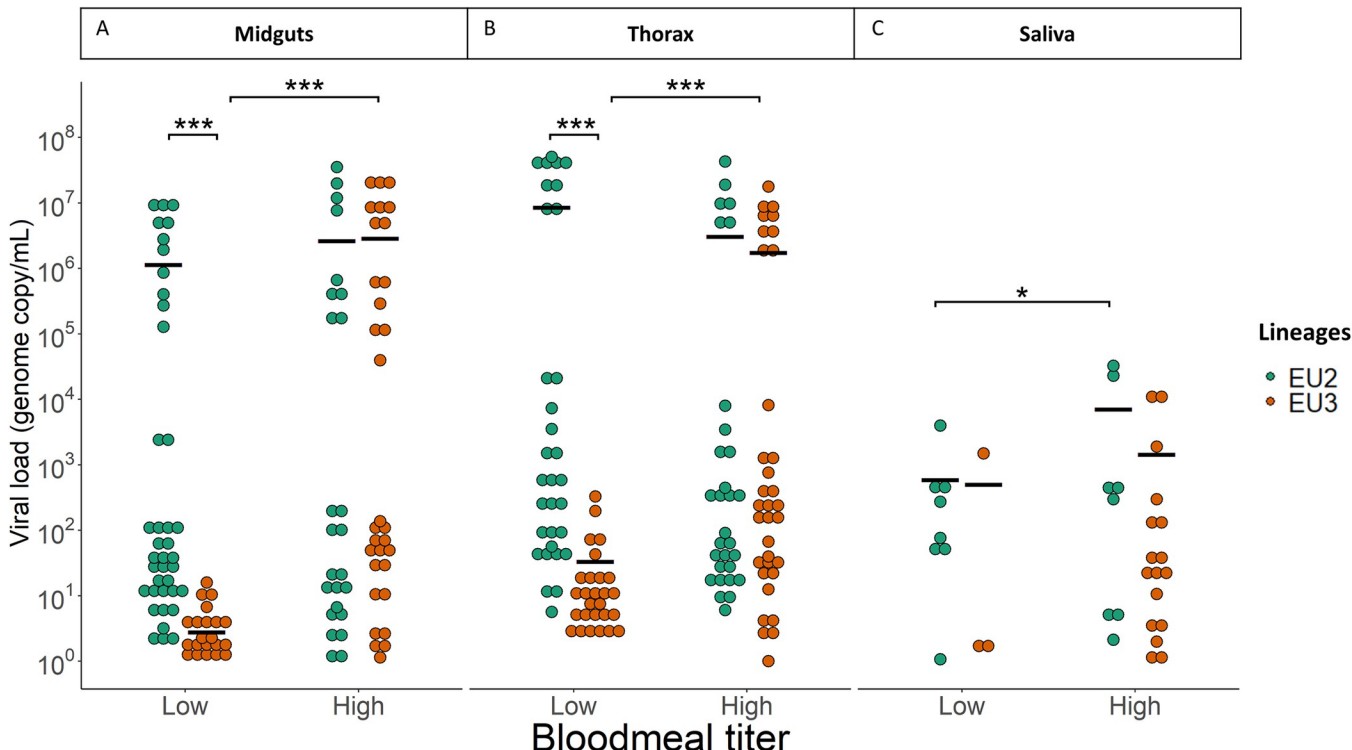

**Fig 3. Viral load of USUV EU2 and EU3 in different *Cx. p. pipiens* tissues.** Viral load (genome copy/μL) is represented for each USUV EU2 (green dots) and EU3 (orange dots) infected individuals, at low and high bloodmeal titers, in A) midguts, B) thorax and C) saliva. The black bar represents the mean. Only significant differences are represented with stars (*: *p*-value < 0.05; ***: *p*-value < 0.001).

showed a significant impact of feeding on infectious blood (*i.e.* being exposed to the virus) compared to feeding on non-infectious blood (Log-rank test: High: $\chi^2_{df=2}$ = 20.3; p-value < 0.0001; Low: $\chi^2_{df=2}$ = 25.1; p-value < 0.0001, Fig 4). This was true at both low and high titer for EU3 (*p*-value < 0.001 and *p*-value = 0.027, respectively, in comparisons with controls). However, EU2 lineage's impact on survival was only significant when fed a high blood meal titer (*p*-value = 0.115 and *p*-value < 0.0001 at low and high titer, respectively). Interestingly, there were no differences in survival between mosquitoes infected with the EU2 and EU3 lineages at high titer (*p*-value = 1), but at low titers, feeding on blood with EU3 lineage had a greater impact on survival than with EU2 (*p*-value = 0.035). Overall, the Hazard Ratios (HR) after feeding on infectious blood were $HR_{low}$ = 7.13 and $HR_{high}$ = 6.81 fold for EU3, and $HR_{low}$ = 2.95 and $HR_{high}$ = 5.8 fold for EU2 (Table 4).

**Table 3. Statistical analyses of USUV EU2 and EU3 viral load in mosquito tissues.**

| | Midgut | | | Thorax | | | Saliva | | |
|---|---|---|---|---|---|---|---|---|---|
| **Variable** | $\chi^2$ | **d.f.** | ***p-value*** | $\chi^2$ | **d.f.** | ***p-value*** | $\chi^2$ | **d.f.** | ***p-value*** |
| **Lineage** | 0.014 | 1 | 0.9 | 0.65 | 1 | 0.42 | 3.10 | 1 | 0.07 |
| **Blood meal titer** | 1.47 | 1 | 0.22 | 2.13 | 1 | 0.14 | **5.88** | **1** | **0.015** |
| **Interaction** | **180.8** | **1** | **<0.0001** | **139.68** | **1** | **<0.0001** | 0.72 | 1 | 0.39 |

Parameters selected in the minimal generalized linear mixed models are presented. $X^2$ is the Chi-square value of the likelihood ratio test and d.f. is the degree of freedom. The p-values are bolded when significant.

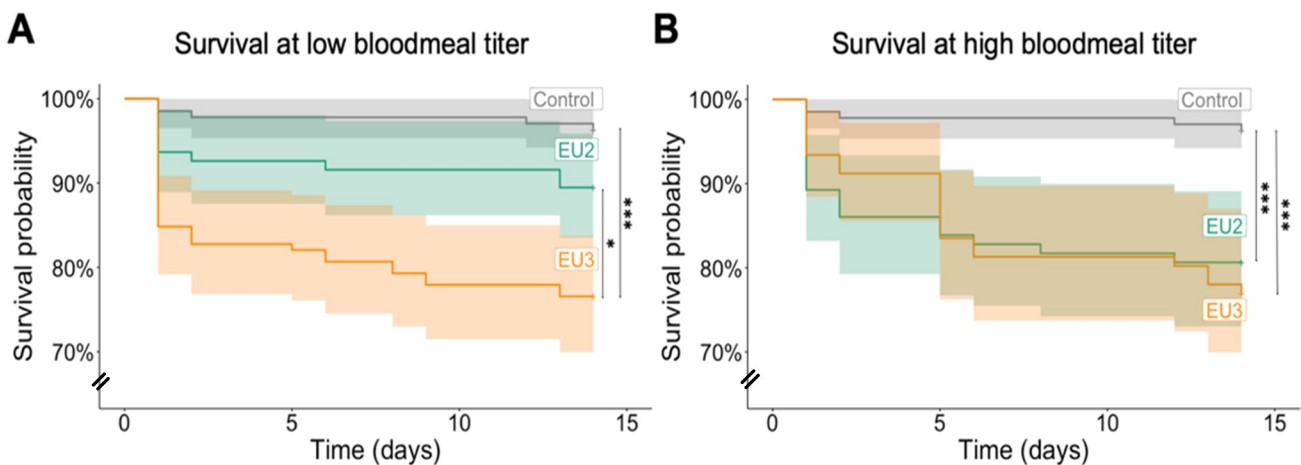

**Fig 4. Effect of infectious blood feeding on *Cx. p. pipiens* survival.** Survival of mosquito blood-fed with USUV EU2 and EU3 at low (A) and at high bloodmeal titer (B) were compared to survival of mosquito fed with non-infectious blood, to assess the combined effects of exposition to and infection by the viruses. All fully blood-fed females have been followed during 14 days and dead females were recorded daily. Pairwise Log-rank tests were used to compare survival between females that fed on infectious versus non-infectious bloodmeal and only significant differences were represented (*: *p*-value < 0.05; ***: *p*-value < 0.001).

In the second analysis, we separated USUV-positive and USUV-negative mosquito midguts for comparisons with controls (Fig 5 and Table 4). We used the infection data obtained on midguts because this is the first place of mosquito-arbovirus interactions that could lead to an effect on mosquito life history traits. By taking into account the infection status we can better test whether replicating virus within mosquitoes, and not just exposure to the virus, impacts survival. For this analysis, we did not include females dead at day one because of the risk of detecting USUV from the blood in their midgut (Sample size: $EU2_{Low}$ = 89 $EU3_{Low}$ = 123 $EU2_{High}$ = 82 $EU3_{High}$ = 82).

At low blood meal titer, the survival of USUV-negative individuals (*i.e.*, no detection EU2 or EU3 viral genome) did not differ from controls (Log-rank test: $\chi^2_{df = 2}$ = 2.1 *p*-value = 0.3; Fig 5A). However, this difference was significant for USUV-positive mosquitoes (Log-rank test: $\chi^2_{df = 2}$ = 7 *p*-value = 0.03; Fig 5B). In pairwise comparisons with controls, the survival of EU2-positive mosquitoes did not differ statistically from controls (EU2 *vs*. CG p-value = 1, $HR_{EU2}$ = 1.89 Fig 5B). In contrast, for EU3-positive mosquitoes there was a significant reduction in survival (EU3 *vs*. CG *p*-value = 0.024, $HR_{EU3}$ = 4.4).

**Table 4. Hazard ratio of USUV lineages associated death with 95% confidence intervals.**

| | | Blood meal titer | | | Blood meal titer | |
|---|---|---|---|---|---|---|
| | | Low | High | Infection status | Low | High |
| USUV lineages | EU2 N = 188 | 2.95 1.0–8.6 | 5.82 2.1–15.7 | Negative N = 92 | 0.58 0.1–4.9 | 2.38 0.6–8.9 |
| | | | | Positive N = 79 | 1.89 0.4–7.9 | 2.32 0.5–9.7 |
| | EU3 N = 236 | 7.13 2.8–18.2 | 6.81 2.5–18.1 | Negative N = 121 | 1.95 0.6–6.4 | 4.65 1.4–15.2 |
| | | | | Positive N = 84 | 4.40 1.3–14.4 | 5.05 1.6–15.5 |

N represents the sample size.

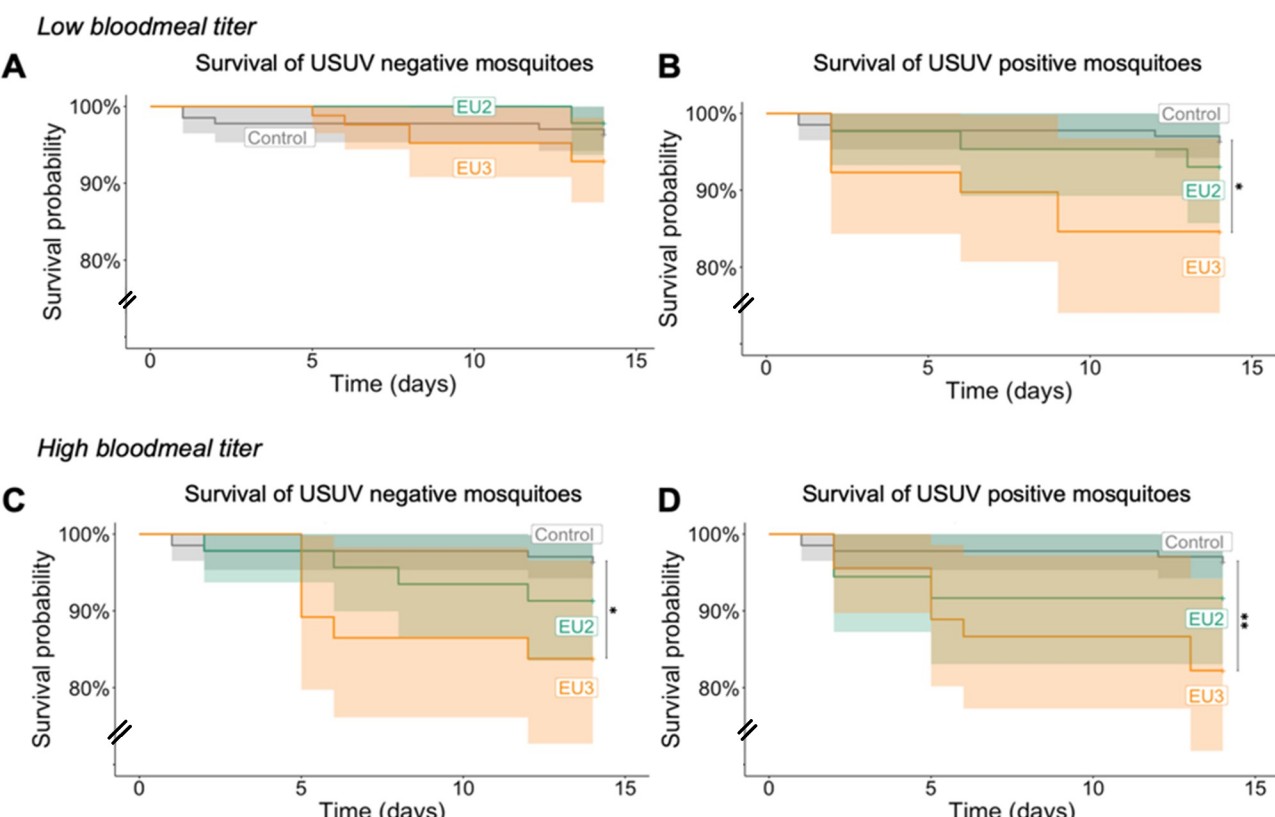

**Fig 5. Effect of USUV EU2 and EU3 infection on *Cx. p. pipiens* survival.** All fully blood-fed females have been followed during 14 days. USUV detection has been performed on dead and surviving mosquitoes except females that died the first day after blood feeding in order to avoid virus detection from undigested blood. The upper panel shows Kaplan Meier survival curves of USUV negative *vs.* control individuals (A) and positive *vs.* control individuals (B) at low bloodmeal titer. The lower panel shows Kaplan Meier survival curves of USUV negative *vs.* control individuals (C) and positive *vs.* control individuals (D) at high bloodmeal titer. Pairwise Log-rank tests were used to compare survival between groups and only significant differences were represented with their p-value. (*: $p$-value < 0.05; **: $p$-value < 0.01).

By contrast, at high blood meal titer, both USUV-negative and USUV-positive individuals showed reduced survival in comparisons with control mosquitoes (Log-rank test: $\chi^2_{df = 2}$ = 7.4 $p$-value = 0.02 and Log-rank test: $\chi^2_{df = 2}$ = 9.7 $p$-value = 0.008, respectively), suggesting that exposure to infectious blood (Fig 5C) and presence of replicating virus (Fig 5D) are both responsible of the observed effect. In pairwise comparisons with controls, taking a blood meal containing a high titer of EU3 lineage significantly reduced survival in both USUV-positive (EU3$_{pos}$ vs. CG $p$-value = 0.0046, HR$_{EU3}$ = 5.03) and negative (EU3$_{neg}$ vs. CG $p$-value = 0.018, HR$_{EU3}$ = 4.64) individuals. By contrast, pairwise comparisons found no significant differences in the survival of USUV-positive or negative individuals that had taken a blood meal with EU2 lineage ($p$-values > 0.05, respectively) despite the significant difference observed when taken as a whole (Fig 4B).

Lastly, we tested whether viral load in USUV-positive mosquitoes affected survival. Cox proportional hazard regression did not show any impact of viral load on survival for either lineage or blood meal titer (Table 5).

## Discussion

Assessing the vectorial capacity for vector-borne diseases is critical to understanding arboviral emergence and characterizing their epidemiological cycles. *Cx. p. pipiens* is a known vector of

**Table 5. Statistical analysis of viral load on mosquito survival.**

| Lineage | Blood meal titer | $\chi^2$ | d.f. | *p-value* |
|---|---|---|---|---|
| EU2 | Low | 0.65 | 1 | 0.421 |
| EU3 | Low | 0.92 | 1 | 0.337 |
| EU2 | High | 0.11 | 1 | 0.739 |
| EU2 | High | 2.00 | 1 | 0.157 |

Cox proportional analyses were performed on infected individuals only. $X^2$ is the Chi-square value of the likelihood ratio test and d.f. is the degree of freedom. The p-values are bolded when significant.

USUV, a virus primarily of birds that has undergone recent phylogenetic and pathogenic divergence as it has spread throughout Europe. While there is known variation among the viral lineages in their virulence and pathogenicity, there has yet been no study of potential among-lineage variation in vector competence in addition to the effect on mosquito survival (as a proxy of virulence). Indeed, as for many arboviruses, vectorial capacity and more specifically vector competence is often tested with only one reference lineage. This neglect of viral diversity of virulence in and transmission from the vector host may profoundly limit our understanding of disease emergence and propagation. This study sought to determine whether these lineages vary in their interaction with the mosquito vector and in the relationship between virulence in the vertebrate host and in the mosquito vector. To better characterize these interactions, we tested two distinct viral titers in blood meal provided to mosquitoes because it influences vector susceptibility to Usutu, West Nile and St Louis encephalitis viruses [22,42,43] as well as virulence [44].

## Vector competence differs between USUV lineages and depends on the blood meal titer

We first examined vector competence by determining viral prevalence of the two lineages in different tissues of *Cx. p. pipiens* (Fig 1). While *Cx. p. pipiens* was competent for both USUV lineages, there was considerable among-replicate and among-individual variability. These differences likely arise from multiple factors, including physiological and immunological response heterogeneity and genetic differences in mosquitoes affecting blood feeding and infection. This variability highlights the need in future studies for adequate sample sizes and replication of artificial blood-feeding assays to generate reliable data on vector competence.

Infection of the mosquito vector proceeds from viral particles entering the midgut following an infectious blood meal, to dissemination into other tissues outside of the midgut, and finally into the saliva where virus can be transmitted when the mosquito takes another blood meal. This within-host viral propagation may be slowed or halted by various barriers to infection (reviewed by Franz et al. [45]). Beyond the physical/cellular barriers, the arthropod immune system may be able at each step to fight the viral infection through many interacting pathways, (e.g., RNA interference (RNAi), JAK-STAT, Toll and Imd pathways [46]). We observed a decrease in USUV prevalence from the midgut, to the thorax, and then the saliva, suggesting barriers do indeed prevent the entrance and propagation of viral particles within mosquito vectors. Our analysis of the infection efficiency and dissemination and transmission rates point to the importance of such barriers and, more interestingly, that the effects of these barriers depend on the viral lineage and the blood meal titer (Fig 2 and Table 2).

The infection efficiency reflects the ability of viral particles to infect the midgut after feeding on infectious blood. Midgut infection barriers prevent viral particles from entering the midgut epithelium due, for instance, to an absence of appropriate surface receptors, as described in

three *Ae. Aegypti* colonies for a Dengue virus strain [47]. In our data about 50% of the mosquitoes that had taken an infectious blood meal had infected midguts, regardless of the differences in blood meal titer, suggesting that the midgut infection barrier is relatively easily overcome. The infection efficiency of the two lineages did not differ significantly (Table 1), suggesting a similar mosquito susceptibility to infection (Fig 1A). Difference in prevalence between blood meal titers was significant when all tissues were analyzed together, suggesting that increasing viral titer in the inoculum may facilitate infection.

From the midgut, viral particles must then move into other tissues. Midgut escape barriers may prevent such dissemination, perhaps through mechanisms acting in the basal lamina of the midgut epithelium [48]. Such barriers will reduce the dissemination rate from the midgut to tissues in the thorax. While overall there was a slight reduction in viral prevalence in mosquito thoraces, there was no effect of viral lineage or titer provided to blood-feeding mosquitoes on the dissemination efficiency, indicating that these lineages are equally efficient in their ability to infect other tissues within mosquito vector, independent of the blood meal titer.

Viral particles must then move into the salivary glands and saliva for transmission to occur when the mosquito vector takes a blood meal. Once again, escape barriers may affect this step in viral propagation by halting or slowing infection of the salivary gland cells or release into the saliva. Because we detected the presence of viral genome in saliva but not infectious particles, our data thus provided evidence of viral replication and release of viral genomes into the saliva allowing to characterize the salivary gland barrier but we cannot conclude on the infectiousness on the saliva. We found that differences in the transmission efficiency and transmission rate of the EU3 lineage depended on the viral titer in the blood meal contrary to EU2 (Figs 1C and 2B and Tables 1 and 2) suggesting that a high titer appears necessary to reach a higher transmission rate for the EU3 lineage but did not for EU2 lineage. This suggests a dose-dependent transmission efficiency for EU3 but not for EU2 lineage: a higher viral load may increase the likelihood of dissemination across the salivary gland barrier for EU3 (Fig 3B and Table 3).

Together, these observations suggest that EU2 replicates more efficiently in mosquitoes than EU3 and that the lineage specific variation to overcome infection barriers and disseminate to the next tissue depend on the inoculating dose of viral particles. Arboviruses may face a proximate viral fitness (defined as the number of viral particles within one host) tradeoff if adaptation to one host environment reduces fitness in the other host [30] due to taxonomic divergence. Our data did not indicate such proximate trade-off as Clé *et al.* [18] described a higher and continuous EU2 replication in human neuronal cells up to six days, while EU3 replication decreased two days post-infection. If similar differences exist in mosquitoes, the less effective EU3 replication could limit within-mosquito viral production when the initial inoculum is low. The viral load in EU2-infected mosquitoes would not be affected by the blood meal titer due in part to its higher replication rate: at high blood meal titer, EU2 viral load may have reached a plateau after only a few days. Infection and replication at earlier and intermediate time points are necessary to test this hypothesis. The poor EU3 replication might also explain the low transmission rate observed at low titer (Fig 3), suggesting that a minimal number of viral particles is required to overcome the salivary gland barrier. This barrier appeared much more difficult to overcome, as both lineages saw a sheer drop in viral load in saliva. It is also worth considering that viral propagation into the salivary glands may occur later in time post-infection. Characterization of the key factors affecting the salivary gland barrier will focus on future genomic and transcriptomic differences between mosquitoes infected by the different lineages and their consequences for vector competence for USUV.

## USUV lineages differ in virulence in vector and vertebrate hosts

We also investigated the virulence of the two USUV lineages in the *Cx. p. pipiens* vector. We first analyzed the effect on mosquito survival of taking an infectious blood meal, which combines the effects of viral exposure and potential infection (viral replication and propagation, Fig 4). To distinguish the impact of infection alone, we then analyzed survival taking into account the infection status of the dead and surviving mosquitoes (Fig 5). Indeed, between 29 to 58% of blood-fed mosquitoes developed an infection (Fig 1A), allowing for an effective comparison of USUV-positive and USUV-negative females (Fig 5).

Overall, it appears EU3 lineage is more virulent to its mosquito host than EU2. Indeed, survival analysis suggests that both exposure to and infection by EU3 affect vector survival. Interestingly, the higher virulence of the EU3 lineage appears little affected by the blood meal titer and despite a lower viral load in mosquitoes (Figs 3 and 4 and Table 4). However, when fed a low blood meal titer, simple exposure to EU3 is not enough to reduce survival (Fig 5A and Table 4). Such reduction in survival could have a great impact on transmission because the daily survival rate is the most influential parameter of the vectorial capacity equation (Eq 1) [49].

The mechanisms promoting EU3's increased virulence in mosquito are not known. It seems likely that it is a consequence of the production of a virulence factor that is present in EU3 but absent in EU2. Such an effect has been reported previously for Dengue, Japanese encephalitis and Zika viruses in *Ae. aegypti*, *Cx. pipiens* and *Ae. albopictus*, where the secretion of NS1 protein in host blood is associated with higher infection and suppression of mosquito immune responses (JAK-STAT and ROS pathways) [50–52] as well as virulence in mammalian hosts [53]. The presence of an EU3-specific NS1 could play a role in the observed differences in virulence. There are three non-synonymous substitutions (R36K, S57T, E139G) among 12 polymorphic sites in the coding region of NS1 between EU2 (MT784899) and EU3 (KX601690) but the functional consequences of these substitutions are not known. In addition, mutations in regulatory non-coding sequences could also be at play. More investigations are required to explore these issues.

An alternative hypothesis is that the reduction in host survival is due to the cost of mounting an immune response. In the frame of the life-history trait theory, trade-offs involving mosquito immune response may arise if the response against pathogens damages host tissues or takes energy and resources away from other fitness-promoting traits such as survival or reproduction. We did not directly measure the USUV-associated immune response in mosquitoes, so we cannot assess the importance of such trade-offs in mosquitoes. The mechanisms of the reduction in host survival, either virus-mediated through the production of virulence factors, host-mediated through the costs of immunity, or a combination of host and viral factors, will require further investigation.

## Different replication, different transmission risks

Epidemiologically, the impact of USUV infection on host survival means less transmission because females will take fewer blood meals—a positive for public health in this context of a newly emerging virus. As the costs of USUV exposition and infection on mosquito vector survival appears different between lineages, it is also expected to differently impact the transmission of these various USUV lineages. To compare the combined effects of viral replication and virulence between the two lineages, we calculated the relative vectorial capacity (Eq 2) using the estimated daily survival and infection/transmission efficiencies (Table 6), the most influential parameters of the vectorial capacity. We hypothesized that EIP does not differ between USUV lineages but it would be interesting to test for variation of EIP between lineages in future studies. We found that EU3 does indeed appear less transmissible at low blood meal

**Table 6.  USUV EU3 over EU2 relative vectorial capacity for *Culex pipiens*.**

| | Blood meal titer | |
|---|---|---|
| | Low | High |
| **Vector infection efficiency** | 0.0013 | 0.0207 |
| **Daily survival rate** | 0.99 | 0.8316 |
| **Relative vectorial capacity (EU3/EU2)** | 0.036 | 0.190 |

titer than EU2 (ratio EU3/EU2 = 0.036), because daily survival rate is relatively reduced; at high blood meal titer, while mosquitoes showed a similar vector competence between EU2 and EU3, the latter remained the less transmissible lineage due to its higher impact on vector survival (ratio EU3/EU2 = 0.19).

From an evolutionary point of view, the reduction in transmission from increased virulence reduces ultimate viral fitness (defined as the number of new infected hosts), resulting in natural selection favoring genotypes that optimize the trade-off between transmission and virulence. This trade-off is generally thought to arise because increasing viral replication has the dual effect of increasing the probability of transmission but also increasing virulence [54]. We found no evidence of such a relationship between neither replication and virulence nor a between replication and transmission in mosquitoes. Indeed, EU3 lineage had lower replication rates, at least at low blood meal titer, but a greater virulence in mosquitoes, the opposite of the expectation from the virulence-transmission trade-off hypothesis. One hypothesis to explain this pattern would be that the immune response against EU3 is deleterious to mosquitoes, leading to earlier death after blood-feeding, but this needs to be confirmed with more data on the extrinsic incubation period and the blood-feeding behavior of mosquito vectors on real vertebrate hosts.

Contrary to what we observed in the mosquito vector, EU2 appears more virulent than EU3 in the vertebrate host [19]. Reduced lifespan of infected vertebrate host would reduce the probability of EU2 transmission to mosquitoes. Although EU2 replicates to a higher level in vertebrates, we showed that prevalence of infection in mosquito midgut were not significantly different between EU2 at high titer and EU3 at low titer. Therefore, while more mosquitoes could be infected by EU3 due to a longer infectious period of the vertebrate hosts, the increased virulence and lower replication rate in mosquito may reduce transmission of EU3 to a new vertebrate host. Therefore, considering the impact of USUV on survival of both vertebrate hosts and mosquito vectors, the propagation of the two lineages should be overall balanced, because of their opposite effect (Fig 6). These results suggest a trade-off between virulence, measured as reduction of survival, and transmission taking into account the whole transmission cycle with both hosts and vectors.

## Conclusion

This study confirms that *Cx. p. pipiens* is a competent vector for USUV and that infection dynamics differ depending on the viral lineage and blood meal titer. At low blood meal titers, EU2 displayed higher viral load and was more often found in saliva, while EU3 is more virulent; the combined effect was that the relative vectorial capacity of mosquitoes infected with EU3 was much lower than with EU2. At high blood meal titers, viral load between the lineages was similar, but the high virulence of EU3 in mosquitoes meant a continued reduction in vectorial capacity. The absence of a relationship between viral load and virulence is inconsistent with the virulence/transmission trade-off. However, our data, coupled with those in a murine model, suggests a possible trade-off between virulence and transmission considering both

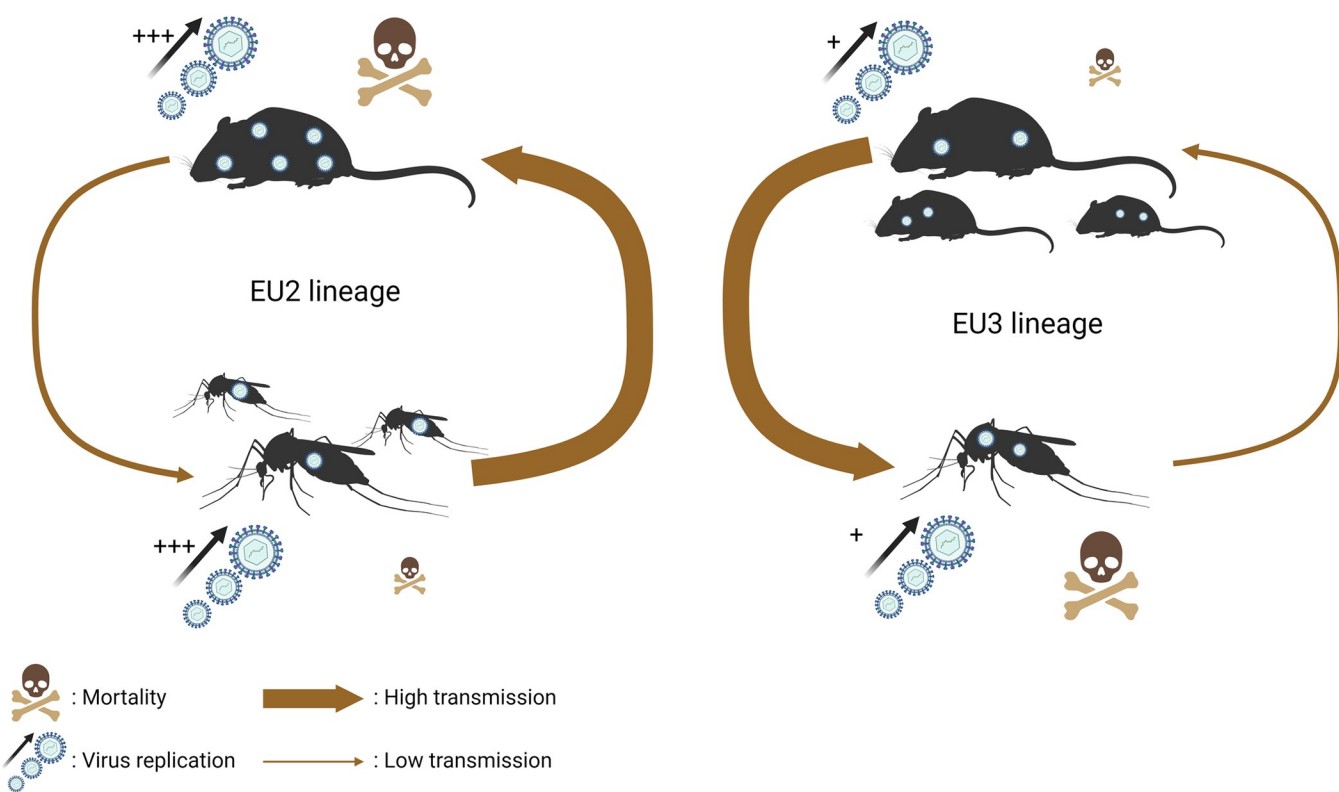

**Fig 6. Schematic synthesis of USUV lineage specific transmission.** The vertebrate (mouse) and arthropods (mosquitoes) hosts are represented on the arbovirus cycle. USUV replication level in hosts are indicated with the (+). The induced mortality rates of the hosts are proportional to the skull size. The transmission intensity is represented by the width of the arrows. The contrasting fitness between the different hosts of each USUV lineage suggests similar propagations of both lineages overall. This figure was created with BioRender.com.

mosquito vector and vertebrate host and demonstrate that viral lineages could be maintained in a balanced equilibrium (Fig 6). Further investigation is required to identify the genetic factors that could explain the lineage-specific differences in virulence and vector competence. Lastly, our study stresses that to fully understand the epidemic dynamics of emergent arboviruses, it is necessary to study more than one viral lineage as well as several isolates of each lineage and not focus solely on vertebrate hosts.

## Supporting information

**S1 Table. Statistical analyses of infection prevalence in mosquito 14 days post infection.** (DOCX)

**S1 Fig. Limit of detection (LOD) of USUV genome by RT-qPCR.** Ct values obtained by RT-qPCR (Nikolay et al. 2014 [36]) were plotted against the corresponding viral RNA titers after serial dilutions (orange for EU3 and blue for EU2). Errors bars represent the standard error of the mean. Viral stocks were titrated at $10^{7.2}$ and $10^{6.8}$ $TCID_{50}$/ml for EU2 and EU3, respectively by the method of SpearmanKärber method [34]. RNA was extracted from 150μl of virus stock using the NucleoMag RNA isolation kit (Macherey-Nagel) and eluted in 50μl before 10-fold serial dilution. RT-qPCR of serial dilutions and negative control were performed in duplicate on every plate (N = 8) containing RNA from mosquito tissues. We defined the limit of detection (*i.e.* individuals are considered infected or uninfected above or below this limit, resp.) as the Ct value of the last standard dilution with reproductible detection (*i.e.* all replicates for a

dilution are positives), here 34 [95CI: 33.11; 34.96], which corresponded to the $10^{-7}$ dilution, i. e to 1 $TCID_{50}$/mL or $10^{-3}$ $TCID_{50}$/µL of virus stock.
(TIFF)

## Acknowledgments

All experiments on arthropods have been performed on the Baillarguet insectarium platform (https://doi.org/10.18167/infrastructure/00001), member of the National Infrastructure EMERG'IN and of the Vectopole Sud network (http://www.vectopole-sud.fr/). The Baillarguet insectarium platform is led by the joint units Intertryp (IRD, CIRAD) and ASTRE (CIRAD, INRAE). We thank Y. Simonin (UMR PCCEI) and N. Dheilly (UMR VIRO) for providing EU2 and EU3 virus isolates, F. Munoz for support in data visualization. We also thank the staff from La Tour du Valat for allowing us to collect *Culex pipiens* larvae. We thank Mylène Weill for helpful comments on this manuscript.

## Author Contributions

**Conceptualization:** Haoues Alout.

**Data curation:** Maxime Prat, Haoues Alout.

**Formal analysis:** Maxime Prat, Haoues Alout.

**Funding acquisition:** Pierrick Labbé, Haoues Alout.

**Investigation:** Maxime Prat, Haoues Alout.

**Methodology:** Maxime Prat, Mélanie Jeanneau, Haoues Alout.

**Project administration:** Pierrick Labbé.

**Resources:** Maxime Prat, Ignace Rakotoarivony, Maxime Duhayon, Yannick Simonin, Giovanni Savini.

**Supervision:** Haoues Alout.

**Visualization:** Maxime Prat.

**Writing – original draft:** Maxime Prat, Haoues Alout.

**Writing – review & editing:** Maxime Prat, Yannick Simonin, Pierrick Labbé, Haoues Alout.

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
