## [Decision Letter · Decision Letter 0]

6 Mar 2024

Dear Dr Alout,

Thank you very much for submitting your manuscript "Virulence and transmission vary between Usutu virus lineages in Culex pipiens" for consideration at PLOS Neglected Tropical Diseases. As with all papers reviewed by the journal, your manuscript was reviewed by members of the editorial board and by several independent reviewers. In light of the reviews (below this email), we would like to invite the resubmission of a significantly-revised version that takes into account the reviewers' comments. 

We cannot make any decision about publication until we have seen the revised manuscript and your response to the reviewers' comments. Your revised manuscript is also likely to be sent to reviewers for further evaluation.

Sincerely,

Mariangela Bonizzoni

Academic Editor

Mabel Carabali

Section Editor

Reviewer's Responses to Questions

**Key Review Criteria Required for Acceptance?**

**Methods**

-Are the objectives of the study clearly articulated with a clear testable hypothesis stated?

-Is the study design appropriate to address the stated objectives?

-Is the population clearly described and appropriate for the hypothesis being tested?

-Is the sample size sufficient to ensure adequate power to address the hypothesis being tested?

-Were correct statistical analysis used to support conclusions?

-Are there concerns about ethical or regulatory requirements being met?

Reviewer #1: The objectives of this study are clearly articulated. The authors did 3 replicates/condition (for a total of 12 independent experiments) and structured well the statistical analysis. The number of mosquitoes tested is appropriate. Ethical and regulatory requirements have been respected. 

I have a major concern regarding the experimental set up. Why the authors choose only one time point for their analysis (14 dpi)? The choice of such long extrinsic incubation period (EIP) poses some problem: what happened before that period? Are there any difference in EIP between the strain? Considering that the authors are looking for different infectivity of the tested strains in mosquitoes, multiple time points should have been tested in order to understand the real fitness of both viral strain. A shorter EIP of one of the two strain can have a big impact in the vectorial capacity. The addition of earlier time points are also supported by the results obtained in the analysis of mortality. Considering that you are using 5-8 days old mosquitoes + 14 days of EIP = 19-22 days of survival for the vector, which under natural condition plus the additional mortality caused by USUV, is not neglectable. I suggest ,if possible, to repeat the experiment adding multiple time points (ie 3dpi (early) and 7dpi (intermediate).

The authors give a general introduction about the different USUV lineages and some more details about EU2 and EU3 (Line 81-83, reference is missing). However, the authors should have include some more detail about current circulation and incidence of this two lineages in Europe. This could have justify better the choice of this two lineages, beside the different virulence. 

The authors used RT-qPCR to quantify the virus. This approach is overall good but careful should be used. Normally other techniques, such as plaque assay or focus forming assay, should be used to assess vector competence in order to really calculate the number of infectious viral particle. Detection of viral genome is not synonyms of infectious viral particles. In the case of TE (saliva sample) where the viral titer is generally low this difference is even more important. If possible i suggest to test at least positive samples by PFA or similar.

The survival data are very interesting, but the data are not well explained. I suggest to rewrite the description, clearly stating what type of sample (live/dead mosquitoes, which day have been excluded etc etc) was used in each graph. The figure quality should be improved. The colored area is not clear and overlapping colors make it difficult to understand. The legend should be improved as well (color code, area meaning).

Reviewer #2: See summary

Reviewer #3: (No Response)

**Results**

-Does the analysis presented match the analysis plan?

-Are the results clearly and completely presented?

-Are the figures (Tables, Images) of sufficient quality for clarity?

Reviewer #1: The authors claim to have tested vectorial capacity multiple times throughout the manuscript. However, by analyzing Infection, dissemination, and transmission, they assessed vector competence. This should be modified throughout the manuscript. 

In lines 466-468, the authors calculate the vectorial capacity. However, it is not clear how they did it. No information about vector density and daily host biting ratio is present in the manuscript, nor have they explained how they calculated/estimated the other parameters in Table 6 (or i don't understand it). This aspect should be clearly reflected in the Materials and Methods and Results sections. Personally, I don’t think the authors should try to calculate vectorial capacity but they should focused only vector competence and the effect of the two tested strain in the mortality. 

The authors frequently use tendency in the description of the results (ie line 225-229). But in my opinion, if a result is not statistically significant speaking about tendency means nothing (this is why we use statistical analysis). i suggest to not use tendency to avoid confusion. Limit the results to what is really significant.

Tables are well designed but i suggest to move Table 4 (nothing is statistically significant) in supplementary material (Table 3 and table 4).

I find the discussion to long, mainly because of systematic repetition of results. I suggest to delete unnecessary repetition or move it to appropriate chapter: ie: introduction (Line 345-349) or Results (Line 400-403; 431-436).

Line 361: The experiment was done under controlled environmental conditions, so how can this be a differential factor?

Line 400-403: Although there is a high titer in the thorax (and midgut), this does not translate into a greater dissemination capacity, suggesting the absence of a dose-dependent mechanism. Do the authors have any insights into this phenomenon

Line 404-406: This finding contrasts with your dissemination data, where no differences were observed regardless of blood titer usage or viral titer reached in the midgut. Indeed, the midgut escape barrier appears to be unaffected by viral titer but rather depends on the virus's physical ability to cross the basal lamina (see reference 10.1038/s41564-019-0619-y)

Line 412: analysis of early and intermediate EIP should be added to verify this hypothesis

Reviewer #2: See summary

Reviewer #3: Results

1. Not sure if I understand the following statistics and statements. “IE (midguts), only the “Lineage” variable had a significant effect with a lower prevalence for EU2 than EU3 (χ² = 4.29; p-value = 0.03, Tab. 1 and Fig. 1A) (low titer: IEEU2 = 0.31 ± 0.19 and IEEU3 = 0.44 ± 0.22, p-value > 0.05; high titer: IEEU2 = 0.44 ± 0.21 and IEEU3 = 0.61 ± 0.21, p-value > 0.05; Fig. 1A”

Figure 1A does not show any significant difference in prevalence for either lineage. 

2. Overall, figures 1 and 2 are part of the same data. The authors can merge these figures. The midgut barriers are well known for a lot of mosquito-borne viruses. This is not new information. 

3. How was the prevalence calculated ? Is it a ct value on qpcr ? What’s the cut off ? The data does not make a lot of sense. e.g. low titers for EU3 lineage show around 50% mean midgut prevalence but figure 3 shows all the viral loads below zero. The same applies to thorax and saliva. 

4. The comparisons in the graphs are not clear. What is the question here ? Is it to compare different lineages or high or low titers ? Just because there is significant difference between something doesn’t mean it has to be shown in the figure. 

5. What is the scale on Y axis in Figure 3 ? How do you have 10-4 genome copies per microliter ? A lot of discussion and focus is on this result. I am not sure if the scale makes sense. The limit of detection of qpcr is never discussed. Anything below 1 genome copy should be considered negative. All the stats need to be redone after removing values below 1 genome copy. I believe this will change all the results and discussion. 

6. I am not sure if I understand the two methods of survival curve. Was every mosquito tested for infection status ? Even the dead ones every day ? How accurate is virus detection in dead mosquitoes ? 

7. Also, the survival curves are a bit misleading. The axis starts at 70% giving an impression of huge difference in survival. Even though it is statistically significant, not sure if 80% vs 100% survival has biological implications. If there are there are, it is not discussed very well. 

8. Why did author use only one time point. Is it possible that the virus can eventually reach the same titers. Only one time point does not provide sufficient information about all the different parameters authors have calculated. Multiple time points in relevant tissues will provide a much clearer picture of viral dynamics within the vector.

**Conclusions**

-Are the conclusions supported by the data presented?

-Are the limitations of analysis clearly described?

-Do the authors discuss how these data can be helpful to advance our understanding of the topic under study?

-Is public health relevance addressed?

Reviewer #1: The conclusions support the data presented and the authors discussed. However the article lack of a limitation chapter in which for exeample the authors can discuss the lack of early and intermediate EIP in their analysis. I am skeptical about the vectorial capacity calculation (see comment above).

Reviewer #2: see summary

Reviewer #3: Discussion

1. It seems like the authors are not sure about their own data. In the first section about vector competence, authors highlight the need to generate reliable data. 

2. It is obvious that the higher virus titer produced better infection in mosquitoes. The rationale behind using these two titers is not clearly discussed. What does it represent biologically ?

3. Line 383, if it’s not statistically significant it shouldn’t be discussed. 

4. Not sure If I understand the following statement “Differences between blood meal titers were not significant but increased slightly at high blood meal titer”

**Editorial and Data Presentation Modifications?**

Reviewer #1: (No Response)

Reviewer #2: - Mosquitoes’ death has been monitored daily for 14 days and dead mosquitoes have been subject to virus detection by RT-qPCR. These data are included in the survival study. However, it would be interesting to look at a possible correlation between the viral load and the time of death in these early dead mosquitoes. This could feed the final model where EU2 lineage has a low virulence while EU3 has a high virulence in mosquitoes and strongly affect their survival.

- The results observed with the viral load (Fig 3) in the different tissues where at low titres EU3 load in the midguts and thoraces is quite low compared to EU2 and at high titres is the same level than EU2 don’t support a model (Fig 6) where EU3 lineage would replicate as much as EU2. Indeed, the authors state in the discussion (line 404) that “these observations suggest that EU2 replicates more efficiently in mosquitoes than EU3”. The model in Fig 6 should be modified (virus +++ > Virus + for EU3 in mosquitoes). 

- In Fig 3 it seems that for EU2 high titres n=29 midguts while n=30 for thoraces. Authors should have recovered the same number of samples for these 2 conditions at max.

- Include the sample size in the appropriate figures.

- Duplicated ref 3 – ref 20

- p20 l439: mere / more

Reviewer #3: Abstract 

1. Shorten the first sentence of the abstract or break into two different sentences. 

2. What is USUV ? Describe the abbreviation once before using it. 

3. What does the following sentence imply ? Infection rate was variable among blood feeding assays but was similar between EU2 and EU3 lineages.

4. There seems to be a lot of confusion in the abstract. Please simplify the statements. In the abstract authors say “ EU2 viral load was high at all titers which suggest a greater replication”. Then in the next line “EU3 was more virulent to mosquitoes than EU2”. Is this referring to mosquito mortality ?

5. When authors mention transmission is it bird-bird transmission ? Are humans considered the dead-end host? 

Introduction

1. Line 67. Too long. Divide into two sentences. Line 67 reads like authors are talking about human outbreaks but the following line mentions outbreaks in birds. 

2. Line 77 reference ?

3. Line 83. Significant difference ?

4. In a multi-host transmission setting what is the daily host biting rate ? Is it for birds or for humans ?

5. Did you study vector competence or transmission of the virus ? Please make sure the right terms are used.

**Summary and General Comments**

Reviewer #1: Additional comment to the Authors:

I suggest condensing lines 87-102. This lengthy paragraph merely provides a general overview of several aspect of vector biology, which is likely familiar to the reader. Instead, I recommend focusing on providing more detailed information on other aspects more relevant to your study, such as elaborating on EU2 and EU3 (including their circulation, detection in vector and host, and outbreaks), as well as explaining the rationale for their inclusion in your research.

Minor suggestion:

Line 62: add ; after family

Line 63: do not capitalize virus

Line 71: characterized instead of typified

Line 77: mutation also facilitate adaptation to vector and/or host

Line 134: if possible add GPS coordinat

---

## [Decision Letter · Decision Letter 1]

4 Jun 2024

Dear Dr Alout,

Thank you very much for submitting your manuscript "Virulence and transmission vary between Usutu virus lineages in Culex pipiens" for consideration at PLOS Neglected Tropical Diseases. As with all papers reviewed by the journal, your manuscript was reviewed by members of the editorial board and by several independent reviewers. The reviewers appreciated the attention to an important topic. Based on the reviews, we are likely to accept this manuscript for publication, providing that you modify the manuscript according to the review recommendations. 

Sincerely,

Mariangela Bonizzoni

Academic Editor

Mabel Carabali

Section Editor

Reviewer's Responses to Questions

**Key Review Criteria Required for Acceptance?**

**Methods**

-Are the objectives of the study clearly articulated with a clear testable hypothesis stated?

-Is the study design appropriate to address the stated objectives?

-Is the population clearly described and appropriate for the hypothesis being tested?

-Is the sample size sufficient to ensure adequate power to address the hypothesis being tested?

-Were correct statistical analysis used to support conclusions?

-Are there concerns about ethical or regulatory requirements being met?

Reviewer #1: -If the author performed the experiments in climatic chamber under controlled environmental conditions (which I understand have not been changed), those cannot be consider variable. Please remove it from the text e.i. line 395 of discussion and adress the discussion.

Reviewer #2: OK

Reviewer #3: (No Response)

**Results**

-Does the analysis presented match the analysis plan?

-Are the results clearly and completely presented?

-Are the figures (Tables, Images) of sufficient quality for clarity?

Reviewer #1: -While your understanding of p-values and the interpretation of statistical significance is correct, discussing tendencies without statistical significance can be misleading and can generate snowball effect mistake when citing (we often see a tendency became significant when cited). Scientific reporting should prioritize clarity and accuracy. Therefore, I recommend that you refrain from discussing tendencies unless they are supported by statistically significant results. This approach will ensure that the conclusions drawn from your study are based on reliable and reproducible data. e.g. remove lines 415-417

-While the challenges of conducting experiments in a BSL3 laboratory are acknowledged, the inclusion of multiple time points is crucial for a comprehensive understanding of viral dynamics and EIP. The current setup precludes a full analysis of these factors. I appreciate the suggestion of the Author to test early time in the discussion (line 512-513) however the manuscript should include a detailed discussion of this limitation.

Reviewer #2: OK

Reviewer #3: (No Response)

**Conclusions**

-Are the conclusions supported by the data presented?

-Are the limitations of analysis clearly described?

-Do the authors discuss how these data can be helpful to advance our understanding of the topic under study?

-Is public health relevance addressed?

Reviewer #1: Now that relative vectorial capacity has been used by Authors and the assumption at the base of their calculation are fully discosed i am more confident with their results.

The study has several limitations (many of them addressed in the reviewed version). I suggest to add a limitation chapter where discuss all of them in a ordinated manner.

Reviewer #2: OK

Reviewer #3: (No Response)

**Editorial and Data Presentation Modifications?**

Reviewer #1: (No Response)

Reviewer #2: None

Reviewer #3: (No Response)

**Summary and General Comments**

Reviewer #1: -The distinction between vector competence and vectorial capacity is crucial and should be meticulously maintained throughout the manuscript. Clear differentiation will prevent any potential misinterpretations by readers. I appreciate the effort of the Authors to clarify this matter in the manuscript and to use a relative vectorial capacity, highlight the assumption of the survival rate and biting rate.

Reviewer #2: The authors well took into account my suggested modifications.

Reviewer #3: (No Response)

PLOS authors have the option to publish the peer review history of their article (what does this mean?). If published, this will include your full peer review and any attached files.

Reviewer #1: No

Reviewer #2: No

Reviewer #3: No

Figure Files:

Data Requirements:

Reproducibility:

References

---

## [Editor Report · Decision Letter 2]

14 Jun 2024

Dear Dr Alout,

We are pleased to inform you that your manuscript 'Virulence and transmission vary between Usutu virus lineages in Culex pipiens' has been provisionally accepted for publication in PLOS Neglected Tropical Diseases.

Best regards,

Mariangela Bonizzoni

Academic Editor

Mabel Carabali

Section Editor

---

## [Editor Report · Acceptance letter]

24 Jun 2024

Dear Dr Alout,

We are delighted to inform you that your manuscript, "Virulence and transmission vary between Usutu virus lineages in Culex pipiens," has been formally accepted for publication in PLOS Neglected Tropical Diseases.

Best regards,

Shaden Kamhawi

co-Editor-in-Chief

Paul Brindley

co-Editor-in-Chief
